# Precision engineering of biological function with large-scale measurements and machine learning

**Drew S. Tack**[1], **Peter D. Tonner**[1], **Abe Pressman**[1], **Nathan D. Olson**[1], **Sasha F. Levy**[2,3], **Eugenia F. Romantseva**[1], **Nina Alperovich**[1], **Olga Vasilyeva**[1], **David Ross**[1]*

**1** National Institute of Standards and Technology, Gaithersburg, MD, United States of America, **2** SLAC National Accelerator Laboratory, Menlo Park, CA, United States of America, **3** Joint Initiative for Metrology in Biology, Stanford, CA, United States of America

* david.ross@nist.gov

**Data Availability Statement:** All relevant data are within the paper and its Supporting Information files.

## Abstract

As synthetic biology expands and accelerates into real-world applications, methods for quantitatively and precisely engineering biological function become increasingly relevant. This is particularly true for applications that require programmed sensing to dynamically regulate gene expression in response to stimuli. However, few methods have been described that can engineer biological sensing with any level of quantitative precision. Here, we present two complementary methods for precision engineering of genetic sensors: *in silico* selection and machine-learning-enabled forward engineering. Both methods use a large-scale genotype-phenotype dataset to identify DNA sequences that encode sensors with quantitatively specified dose response. First, we show that *in silico* selection can be used to engineer sensors with a wide range of dose-response curves. To demonstrate *in silico* selection for precise, multi-objective engineering, we simultaneously tune a genetic sensor's sensitivity ($EC_{50}$) and saturating output to meet quantitative specifications. In addition, we engineer sensors with inverted dose-response and specified $EC_{50}$. Second, we demonstrate a machine-learning-enabled approach to predictively engineer genetic sensors with mutation combinations that are not present in the large-scale dataset. We show that the interpretable machine learning results can be combined with a biophysical model to engineer sensors with improved inverted dose-response curves.

## Introduction

As the field of synthetic biology transitions from a qualitative, trial-and-error endeavor into a mature engineering discipline, methods that enable the engineering of biological function with quantitative precision are required, i.e., to produce an outcome that meets a quantitative specification. This need is particularly acute for genetic sensors, which form the basis for synthetic gene circuits and related approaches for programming cells to regulate gene expression dynamically in response to environmental stimuli.

**Funding:** HHS | National Institutes of Health (NIH):
Sasha F Levy R01 HG011676; HHS | National
Institutes of Health (NIH):Sasha F Levy R01
AI164530. The funders had no role in study design,
data collection and analysis, decision to publish, or
preparation of the manuscript.

**Competing interests:** The authors declare that they
have no conflict of interest.

Most efforts to engineer genetic sensors have been qualitative in nature, e.g., demonstrations of new sensor architectures or sensors that respond to new inputs [1–6]. Those qualitative demonstrations are the necessary first steps in developing a toolkit of sensors for synthetic biology and for demonstrating the variety of cellular control circuits enabled by genetic sensors. However, for many applications, genetic sensors will also need to be engineered with a quantitatively specified dose-response curve matched to each application [2,4,7–10]. That dose-response curve is typically described using a version of the Hill equation:

$$G(L) = G_0 + \frac{G_\infty - G_0}{1 + \left(\frac{EC_{50}}{L}\right)^n},$$

where $L$ is the input signal level (e.g., concentration of ligand); $G(L)$ is the regulated gene expression output from the sensor as a function of the input signal; $G_0$ is the basal output level; $G_\infty$ is the saturating output level; $EC_{50}$ is the input level required to give an output midway between $G_0$ and $G_\infty$; and $n$ is the Hill coefficient, which quantifies the steepness of the dose response.

Although the importance of tuning the dose response of genetic sensors has been recognized for applications such as engineered living therapeutics, dynamic pathway control, and enzyme engineering [2,4,7–9,11,12], very few methods have been described that can accomplish the required tuning with any level of quantitative precision or accuracy. With RNA-based genetic sensors (e.g., riboswitches), the relatively predictable biophysics of base-pair interactions has enabled methods to engineer new sensors with quantitatively predictable $G_0$ and $G_\infty$ [13,14]. For protein-based genetic sensors, general guidelines have been given for tuning dose-response curves [7,10,15,16], and several methods have been demonstrated to improve sensor performance by reducing $EC_{50}$ or increasing the dynamic range ($G_\infty/G_0$) [17–26]. But no methods have yet been described that can engineer protein-based sensors with specific quantitative values for the parameters of the Hill equation.

Here, we leverage a large-scale, genotype-phenotype dataset to demonstrate two methods for quantitatively precise engineering of protein-based genetic sensors: *in silico* selection, and forward engineering enabled by machine-learning (ML). With *in silico* selection, we mine the large-scale dataset to find DNA sequences that encode genetic sensors that meet quantitative specifications. We show that *in silico* selection can be used to engineer genetic sensors with $EC_{50}$ values spanning a wide range (from 3 μmol/L to over 1000 μmol/L) and with quantitative accuracy (within about 1.3-fold). In addition, we demonstrate *in silico* selection for precise, multi-objective engineering: first, by engineering genetic sensors with both $EC_{50}$ and $G_\infty$ within about 1.2-fold of specified values; and second, by engineering sensors with inverted dose-response and $EC_{50}$ within about 2-fold of specified values. With ML-enabled forward engineering, we use the large-scale dataset to train an interpretable ML model, and we show that the model can predict both $EC_{50}$ and $G_\infty$ for novel combinations of mutations, also with high accuracy (within 1.9-fold and 1.2-fold for $EC_{50}$ and $G_\infty$, respectively). Finally, we use results from the interpretable ML model in combination with guidance from a biophysical model, to engineer new inverted LacI variants with improved $EC_{50}$ and $G_\infty$.

## Results

Many previous publications have described the effects of protein mutations on genetic sensor dose-response curves. However, we are not aware of any previous work that has demonstrated the use of protein mutations to tune a genetic sensor dose-response curve to meet quantitative specifications. So, the objectives of this manuscript are to demonstrate methods whereby protein mutations can be used for quantitative tuning of dose-response curves and to test the

accuracy and precision of those methods. To that end, the primary statistic we will use to assess different methods is the fold-accuracy: $\exp[\text{RMSE}(\ln(x))]$, where $x$ is the parameter to be tuned (e.g., $EC_{50}$, $G_{\infty}$ from the Hill equation), and $\text{RMSE}(\ln(x))$ is the root-mean-square difference between the logarithm of the actual value of $x$ and the logarithm of the targeted or predicted value of $x$. We use the logarithmic scale to assess accuracy because the parameters of a genetic sensor dose-response curve can span multiple orders of magnitude and because the resulting fold-accuracy is the most suitable metric for applications of engineered genetic regulatory networks [27].

The methods we demonstrate here both require a large-scale genotype-phenotype dataset as a starting point (e.g., deep mutational scanning). For that, we used a recently published dataset that contains dose-response curves for over 60,000 variants of a protein-based genetic sensor, the *lac* repressor, LacI [28]. Briefly, to create the large-scale genotype-phenotype dataset, error-prone PCR was used to generate a library of LacI variants with an average of 7.0 DNA mutations and 4.4 missense mutations (i.e., amino acid substitutions) per coding sequence. The library was barcoded and a growth-based barcode counting assay was used to measure the dose-response curve, $G(L)$, for every variant in the library. Each dose-response curve was fit to the Hill equation to provide estimates for the Hill equation parameters and their associated uncertainties. In addition, long-read sequencing was used to measure the full-length protein coding sequence for each barcoded variant.

## Precision engineering via *in silico* selection

The concept of *in silico* selection is fairly simple: use the large-scale dataset as a lookup table to identify variants with desired phenotypes along with their matching genotypes. That information can then be used to synthesize DNA sequences that will result in the required protein phenotype (i.e., dose-response curve). The keys to successful precision engineering with *in silico* selection are the number of measured variants and the diversity of phenotypes spanned by the large-scale dataset. The dataset must include sufficient diversity to cover the range of functional outcomes needed for the engineering objectives. For example, the LacI dataset includes variants with $EC_{50}$ values from less than 1 μmol/L to over 1000 μmol/L (Fig 1). So, with that dataset, it should be possible to engineer LacI variants with a wide range of $EC_{50}$ values. As a first test of the *in silico* selection approach, we used the genotype-phenotype dataset to identify a set of LacI variants with $EC_{50}$ ranging from about 3 μmol/L to over 1000 μmol/L (and with $G_0$ and $G_{\infty}$ near the wild-type values). For each of those variants, we then synthesized the LacI coding sequence, integrated it into a plasmid where it regulated the expression of a fluorescent protein, and measured the resulting *in vivo* dose-response curves using flow cytometry (Fig 2A). The results indicate a fold-accuracy of 1.67 for engineering LacI variants with different $EC_{50}$ values (Fig 2B; where we calculate the fold-accuracy as described above, using $EC_{50}$ reported in the large-scale dataset as the predicted values and $EC_{50}$ determined by flow cytometry as the actual values). However, there is a systematic error between the cytometry measurements and the large-scale dataset: at low $EC_{50}$, the cytometry result tends to be higher than the large-scale result, while at high $EC_{50}$, the cytometry result tends to be lower (Fig 2C). After correcting for this systematic error (using a linear fit to the $\ln(EC_{50})$ data shown in Fig 2B for the predicted values), we calculate a best-case fold-accuracy of 1.31 for *in silico* selection of $EC_{50}$. For the subsequent evaluations of *in silico* selection described below, we continued to apply this correction to identify variants with $EC_{50}$ values satisfying quantitative specifications.

In addition to providing quantitative accuracy and precision for a single phenotypic parameter, *in silico* selection is particularly well suited to multi-objective optimization of protein function. With *in silico* selection, one can simply search the large-scale dataset for sequence

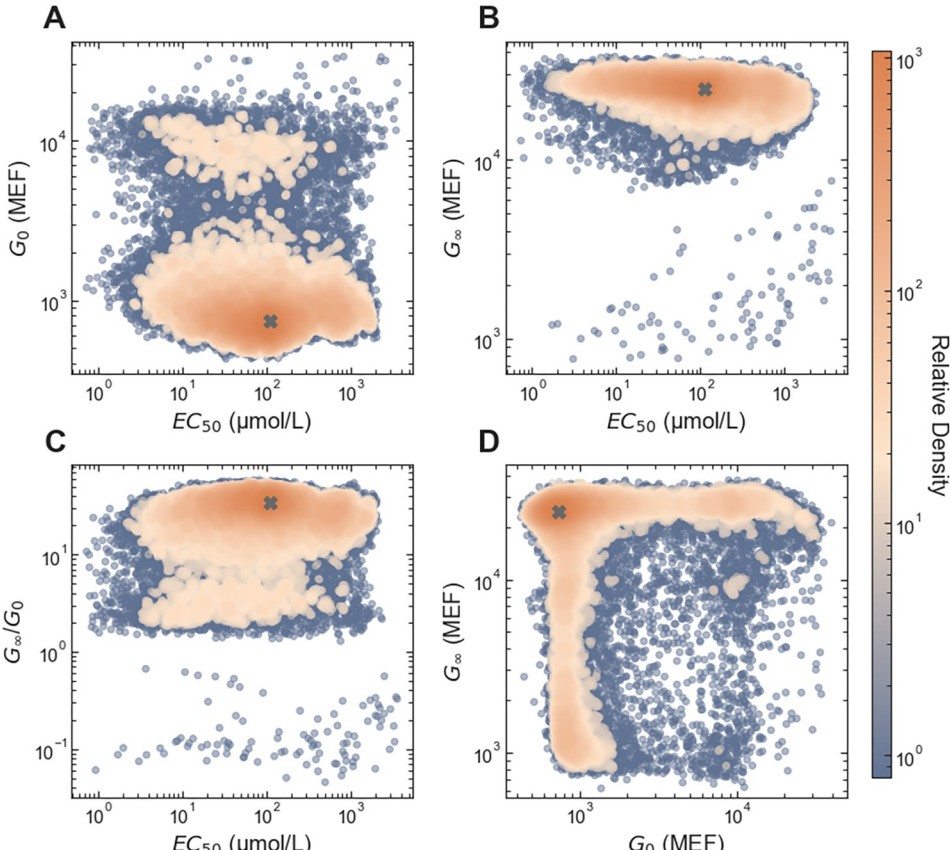

**Fig 1. Diversity of dose-response phenotypes in the large-scale dataset.** The colored points are the values as reported in the genotype-phenotype dataset, with colors indicating the relative density of similar phenotypes. The gray 'X' in each plot shows the parameter values for the wild-type LacI dose-response curve.

variants that satisfy multiple criteria simultaneously. This avoids the need for complicated multi-objective Darwinian selection schemes that are necessary for directed evolution. Both $EC_{50}$ and $G_\infty$ need to be quantitatively tuned for optimal dynamic control of a metabolic pathway using a genetic sensor [9]. So, to demonstrate multi-objective optimization with *in silico* selection, we first defined a set of quantitative specifications for $EC_{50}$ and $G_\infty$. For those specifications, we chose a grid of $EC_{50}$ and $G_\infty$ values with $EC_{50}$ equal to 10 μmol/L, 30 μmol/L, or 100 μmol/L, and with $G_\infty$ equal to 16 kMEF or 25 kMEF (the units, MEF, are molecules of equivalent fluorescein from the calibration of cytometry data with fluorescent beads, see Materials and Methods). Next, we used the large-scale dataset to identify the DNA sequences most likely to encode LacI variants with both $EC_{50}$ and $G_\infty$ close to the specified values (after correcting for the systematic error in $EC_{50}$ as described above). In most cases, we chose the top three sequences for each specification (ranked by the probability of $EC_{50}$ within 1.2-fold and $G_\infty$ within 1.1-fold of the target, based on the large-scale measurement uncertainty). For $EC_{50}$ = 100 μmol/L, $G_\infty$ = 16 kMEF, the top two sequences were very similar (encoding for the missense mutation V95M, plus mutations to the disordered loops near the LacI tetramer helix), so for this specification, we also chose the fourth-ranked sequence. The specification $EC_{50}$ = 100 μmol/L, $G_\infty$ = 25 kMEF is very close to the wild-type LacI phenotype, so we did not choose any sequences for that specification. We then synthesized each sequence, integrated it into a plasmid where it regulated the expression of a fluorescent protein, and measured the resulting

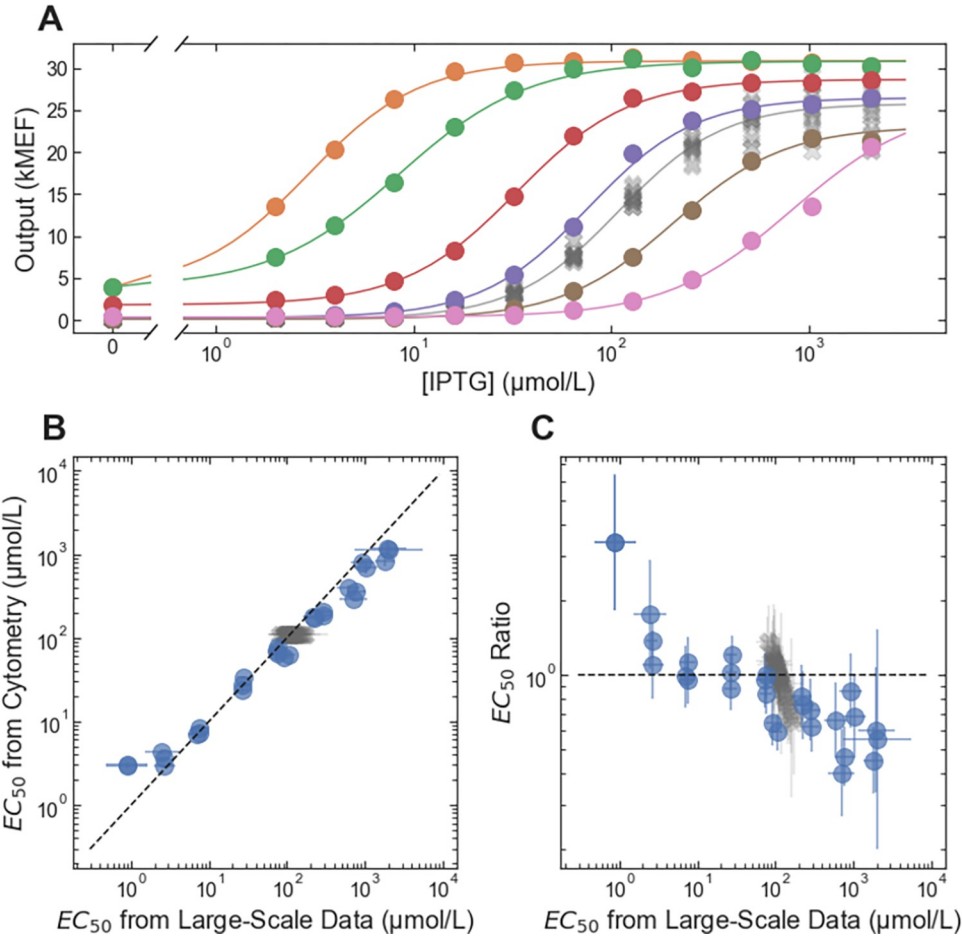

**Fig 2. Accuracy and precision of $EC_{50}$ from *in silico* selection.** (A) Example dose-response curves for LacI variants selected to span a wide range of $EC_{50}$ values. Each variant is plotted with a different color, with lines showing the fits to the dose-response using the Hill equation. The wild-type dose response is plotted with the gray 'X' markers. (B) $EC_{50}$ from the flow cytometry measurements plotted vs. $EC_{50}$ from the large-scale dataset. The dashed line indicates equality between the cytometry and large-scale results. (C) The ratio: ($EC_{50}$ from flow cytometry) ÷ ($EC_{50}$ from the large-scale dataset) plotted vs. $EC_{50}$ from the large-scale dataset. In both B and C, results for non-wild-type LacI variants are plotted with blue circles, and results for wild-type LacI are plotted with gray X's (there were multiple copies of the wild-type in the large-scale dataset, each plotted separately). Error bars indicate ± one standard deviation.

*in vivo* dose-response curves using flow cytometry (Fig 3A). Comparing the cytometry results with the corresponding multi-objective specifications, the *in silico* selection approach showed good performance, with 1.22-fold and 1.14-fold accuracy for $EC_{50}$ and $G_\infty$, respectively. However, there was some systematic deviation from the targeted $G_\infty$ for specifications with $G_\infty$ = 25 kMEF (Fig 3B). Note that the slight apparent improvement in the $EC_{50}$ fold-accuracy compared with the "best-case" fold-accuracy determined above is probably not significant given the finite number of data points used (N = 16).

As a final test of the *in silico* selection approach, we used it to engineer LacI variants with inverted dose-response ($G_\infty < G_0$) and with specified $EC_{50}$. To identify sequences from the large-scale dataset, we used criteria similar to those described above to choose the sequences most likely to encode inverted LacI variants with $EC_{50}$ equal to 10 μmol/L, 30 μmol/L, or 100 μmol/L (applying the $EC_{50}$ correction described above). The dataset contains a much lower density of inverted variants (Fig 1C, $G_\infty/G_0 < 1$). So, for each target specification, there

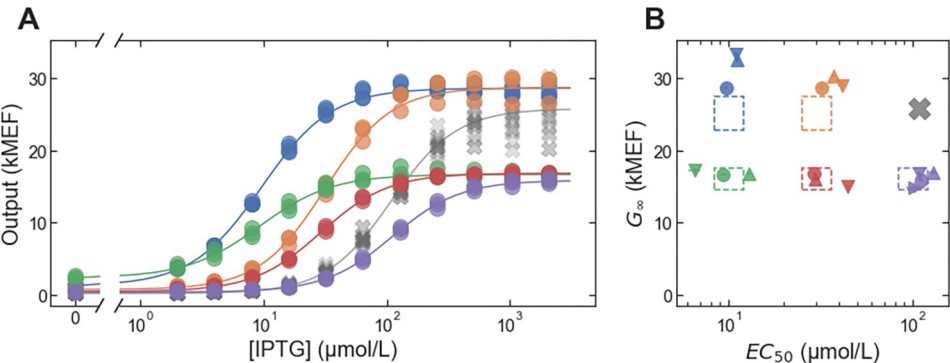

**Fig 3. Multi-objective *in silico* selection of LacI variants with different $EC_{50}$ and $G_\infty$ values.** (A) Example dose-response curves for LacI variants selected to satisfy multi-objective specifications for $EC_{50}$ and $G_\infty$. One variant is plotted for each target specification, each with a different color and with lines showing the fits to the dose-response using the Hill equation. The wild-type dose response is plotted with the gray 'X' markers. (B) Evaluation of multi-objective selection performance. The dashed rectangles show the target specifications in a 2D plot of $G_\infty$ vs. $EC_{50}$, with a different color for each specification. For each specification, three or four distinct LacI variants were selected, and the resulting $G_\infty$ and $EC_{50}$ values (from cytometry) for those variants are plotted with different markers (with marker color indicating the targeted specification). Error bars indicate ± one standard deviation and are typically smaller than the markers.

was only a single sequence with a greater than 20% probability of having an $EC_{50}$ within 1.5-fold of the targeted value (based on the uncertainty of the large-scale results). The sparsity of inverted variants is at least partially due to the FACS pre-screening that was applied before the large-scale measurement to reduce the fraction of variants with high $G_0$ [28], which would have removed all inverted variants from the measured library had it been perfectly efficient.

As before, we synthesized the sequences identified by *in silico* selection, and we measured the *in vivo* dose-response curves for the resulting LacI variants with flow cytometry (Fig 4A). All three variants had inverted dose-response curves with $G_0$ and $G_\infty$ satisfying the targeted specification ($G_0$ within 1.3-fold of 25 kMEF and $G_\infty < 12.5$ kMEF, Fig 4B). However, for each of the sequences, the resulting $EC_{50}$ was higher than the targeted values (by 1.9-fold,

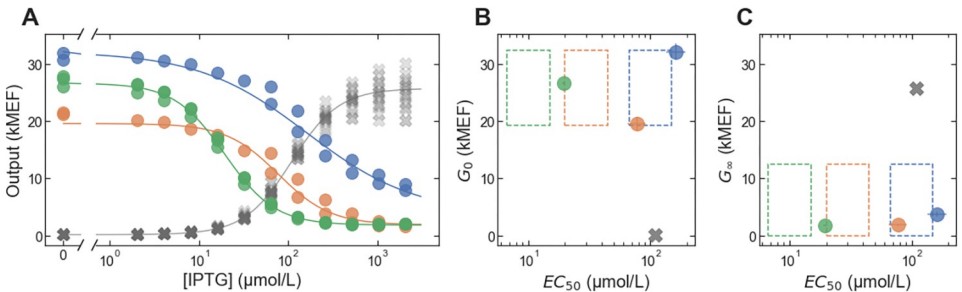

**Fig 4. Multi-objective *in silico* selection of inverted LacI variants.** (A) Dose-response curves for LacI variants selected to have inverted dose-response curves with specified $EC_{50}$. One variant is plotted for each target specification, each with a different color and with lines showing the fits to the dose-response using the Hill equation. The wild-type dose response is plotted with the gray 'X' markers. (B-C) Evaluation of multi-objective selection performance. The dashed rectangles show the target specifications in a 2D plot of $G_0$ (B) and $G_\infty$ (C) vs. $EC_{50}$, each with a different color. For each specification, one LacI variant was selected, and the resulting $G_0$, $G_\infty$ and $EC_{50}$ values (from cytometry) for those variants are plotted (with marker color indicating the targeted specification). For comparison, the wild-type $G_0$, $G_\infty$ and $EC_{50}$ are plotted with gray 'X' markers. Error bars indicate ± one standard deviation and are typically smaller than the markers.

2.6-fold, and 1.6-fold for targeted $EC_{50}$ of 10 μmol/L, 30 μmol/L, and 100 μmol/L, respectively).

To determine whether the deviations from the targeted $EC_{50}$ were due to systematic errors in the large-scale measurement, we synthesized and measured the dose-response for eight additional sequences, selected only based on the inverted phenotype (without a specified $EC_{50}$). The cytometry results confirm that all eight variants have inverted dose-response curves (Fig 5). Furthermore, the results indicate an accuracy of 2.8-fold for $EC_{50}$ of the inverted variants, with no systematic bias (Fig 6). The results in Fig 6A are shown with the $EC_{50}$ correction described above. The lower accuracy for the inverted variants (compared with the results in Fig 2B) is consistent with the estimated uncertainty of the large-scale measurements, and is due to the FACS pre-screening, which reduced the number of barcode reads associated with each inverted variant.

## ML-enabled forward engineering

For some applications, it can be important to predict the phenotype resulting from combinations of mutations that are not present in the large-scale dataset (e.g., to apply sequence constraints that could not be easily applied during construction of the large-scale library). In those situations, the large-scale data can be used to train a machine-learning (ML) models that can then be used to predict the phenotype resulting from novel combinations of mutations. To demonstrate this approach, we used the large-scale LacI dataset to train an ML model using LANTERN, a recently described approach that learns interpretable models of genotype-phenotype landscapes and that also provides good predictive accuracy (e.g., as good or better than neural network models) [29]. Cross validation results for the LANTERN model trained with the LacI dataset are shown in Fig 3 of reference [29]. We used the resulting model to predict $EC_{50}$ and $G_{\infty}$ for 33 variants with mutation combinations that are not found in the large-scale dataset–and using only a restricted set of 16 missense mutations. We chose the 16 mutations to give a range of different effects on the dose-response, and we used mutations distributed across the LacI core domain (Fig 7, S1 Appendix) but avoided mutations to the DNA binding domain that might disrupt interactions between LacI and its cognate DNA operator [21]. We then synthesized the LacI sequences for the 33 variants, measured their dose-response with cytometry, and compared the results with the predictions from the LANTERN model. Overall, the prediction accuracy of the LANTERN model was nearly as good as the accuracy of the underlying measurements, with 1.93-fold and 1.19-fold accuracy for $EC_{50}$ and $G_{\infty}$, respectively (Fig 8).

Surprisingly, five of the 33 variants had inverted dose-response curves, and all five had the same missense mutation: V136E. In addition, two double mutants with the V136E mutation had non-monotonic dose-response: the double mutant V136E/G200C had a band-stop dose-response curve (referred to as the "reversed" phenotype in earlier literature [31–37]); and the double mutant V136E/S279T had a more complicated non-monotonic dose-response (high-low-high-low). We did not include the data for V136E/G200C or V136E/S279T in the quantitative comparison (Fig 8), because it did not match the form of the Hill equation. The single mutation V136E, applied to the wild-type background, gives a dose-response with reduced $G_{\infty}$ but $G_0$ and $EC_{50}$ similar to the wild type (Fig 7). Previous work has shown that single mutations that reduce $G_{\infty}$ relative to the wild-type can be intermediates toward the evolution of the inverted phenotype [38–40], though V136E is located more on the periphery of the protein structure than the intermediate mutations in those previous studies. The prediction accuracy for the five inverted variants was generally poor, particularly for $EC_{50}$. This discrepancy was not surprising: the large-scale dataset used to train the model contained few examples of

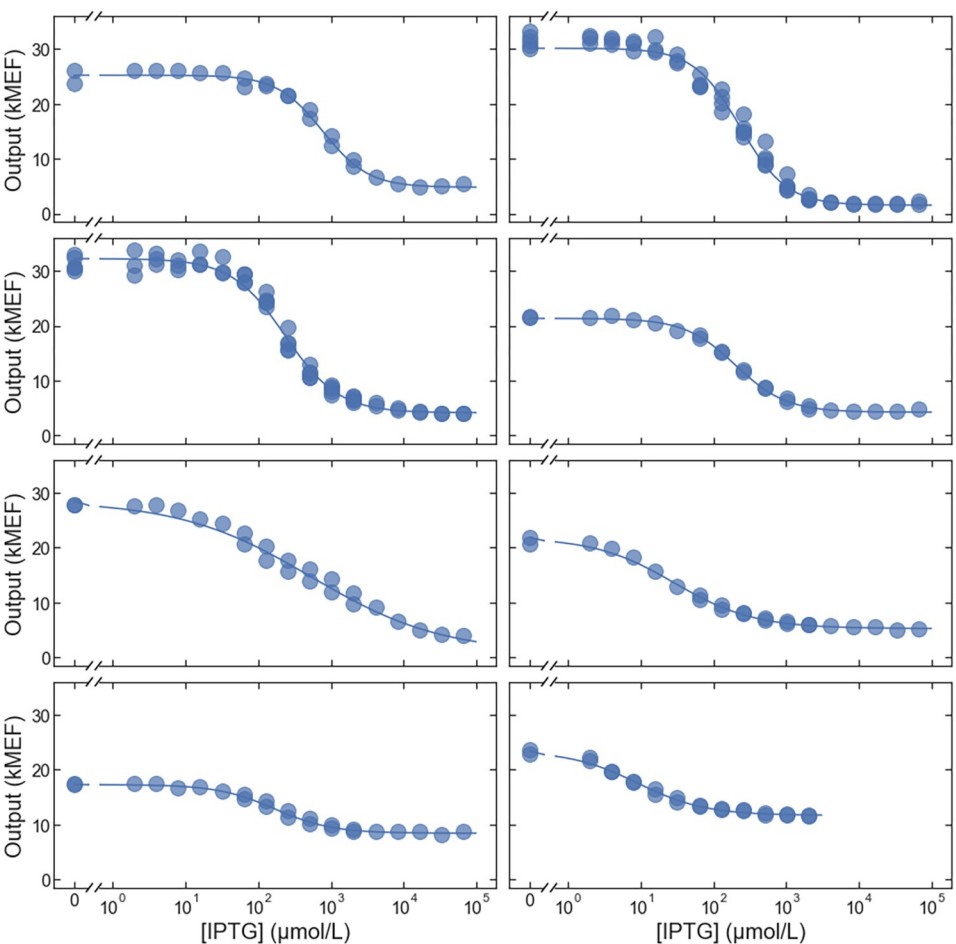

**Fig 5. Additional inverted variants.** Dose-response curves for eight additional inverted LacI variants selected to test the accuracy of the large-scale measurements.

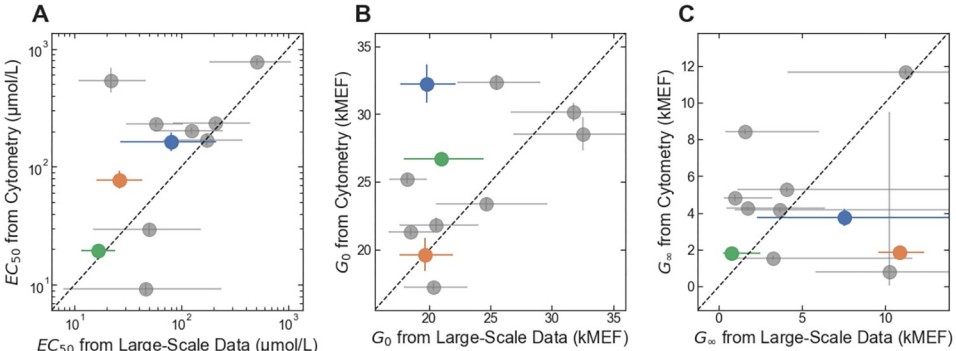

**Fig 6. Accuracy of large-scale measurement for inverted variants.** (A) $EC_{50}$ from the flow cytometry measurements plotted vs. $EC_{50}$ from the large-scale dataset. (B) $G_0$ from the flow cytometry measurements plotted vs. $G_0$ from the large-scale dataset. (C) $G_\infty$ from the flow cytometry measurements plotted vs. $G_\infty$ from the large-scale dataset. In all three plots, results for the inverted variants selected to have specified $EC_{50}$ are plotted with markers colored to match the results in Fig 4; results for additional inverted variants are plotted with gray markers. Error bars indicate ± one standard deviation.

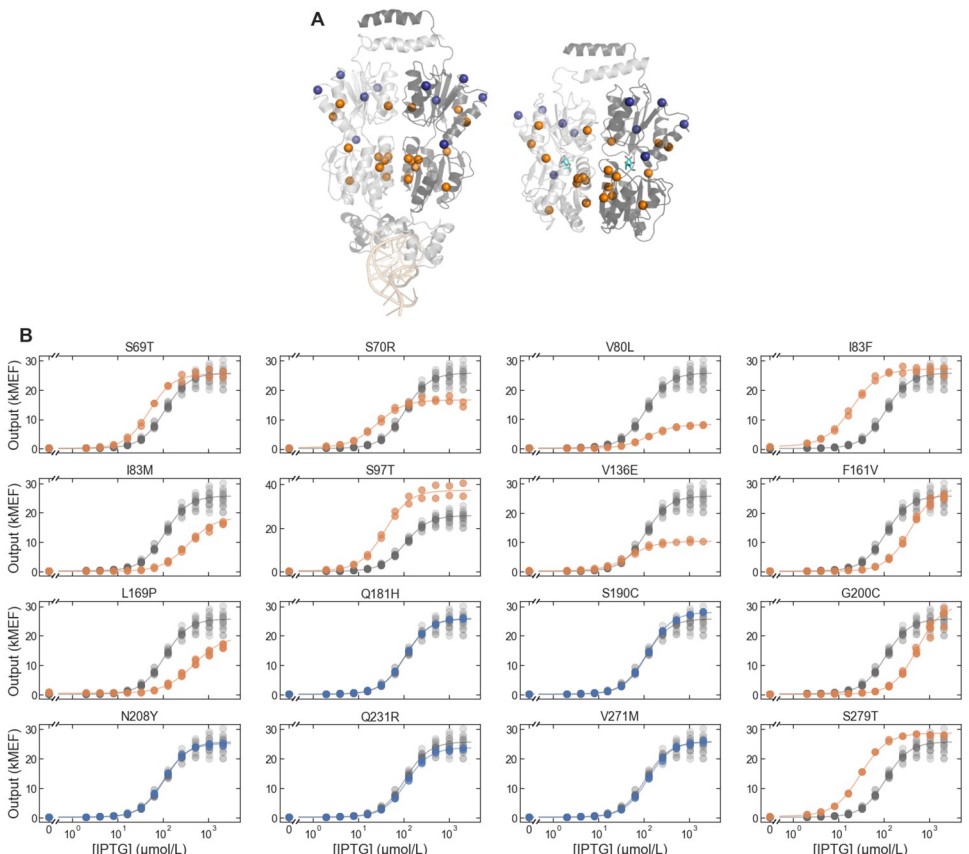

**Fig 7. Mutations used for ML-enabled forward engineering.** (A) LacI protein structure showing location of mutations. The DNA-binding configuration is shown on the left (DNA at the bottom of the structure in light orange, PDB ID: 1LBG [30]) and the ligand-binding configuration is shown on the right (IPTG in cyan, PDB ID: 1LBH [30]). Both configurations are shown with the view oriented along the protein dimer interface, with one monomer in light gray and the other monomer in dark gray. Colored spheres highlight the positions of mutations used for ML-enabled forward engineering, with silent mutations in blue and non-silent mutations in orange. (B) Dose-response of single-mutant LacI variants with each of the mutations used for ML-enabled forward engineering. In each plot, the single-mutant dose-response is plotted in blue (for silent mutations) or orange (for non-silent mutations), and the wild-type dose response is plotted in gray.

inverted variants, and so the model could not learn to predict them. If we consider only the 28 non-inverted variants tested, the prediction accuracy of the LANTERN model improves significantly for $EC_{50}$ (1.31-fold) but only slightly for $G_{\infty}$ (1.17-fold).

In addition to accurately predicting phenotype from genotype, LANTERN learns interpretable models [29]. Part of this interpretability comes from the way LANTERN learns to represent the effect of each mutation. LANTERN represents each mutational effect as a vector in a low dimensional latent space (three dimensions for the LacI dataset), and the combined effect of multiple mutations is simply represented as the sum of the corresponding vectors. The different components of the latent vector space learned by a LANTERN model often resemble a set of latent biophysical parameters (e.g., free energies) that control the protein phenotype. However, the latent parameters learned by a LANTERN model are unlabeled, meaning that while a connection between the parameters learned by LANTERN and biophysical parameters may exist, the model does not identify this connection. But, when an explicit biophysical model is available, it can potentially be linked to the parameters learned by LANTERN. This

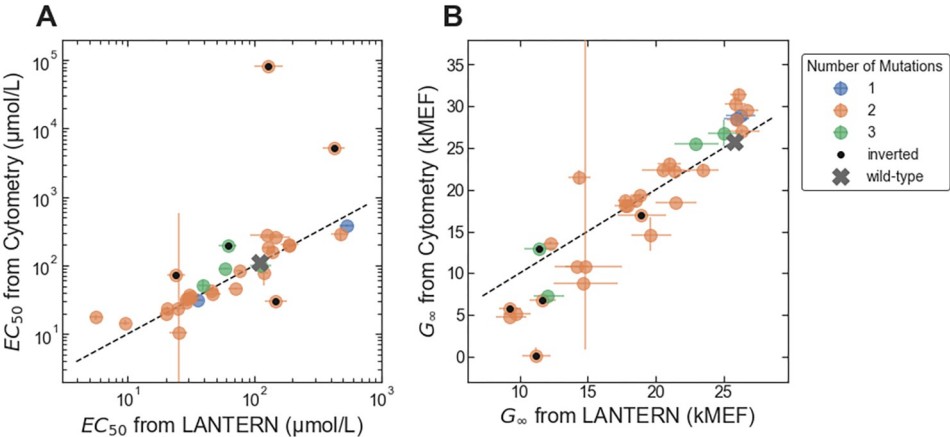

**Fig 8. Accuracy of ML-enabled forward engineering.** (A) $EC_{50}$ from the flow cytometry measurements plotted vs. $EC_{50}$ predicted by the LANTERN ML model. (B) $G_\infty$ from the flow cytometry measurements plotted vs. $G_\infty$ predicted by the LANTERN ML model. In each plot, results for LacI variants with different numbers of mutations are plotted with different colors. Results for the five unexpectedly inverted variants are marked with black dots. Error bars indicate ± one standard deviation.

has been demonstrated qualitatively for a biophysical model of LacI function [41–44] and the LANTERN model trained on the large-scale LacI dataset [29]. More specifically, the first (most significant) latent parameter learned by the LANTERN model seems to correspond to changes to any one of three parameters in the biophysical model (the binding free energy for LacI to its DNA operator, $\Delta\varepsilon_{RA}$; the logarithm of the LacI allosteric constant, $\Delta\varepsilon_{AI}$; or the ligand binding constant for the inactive state of LacI, $K_I$; using the notation of [41,43]). The second latent parameter, however, seems to correspond to changes to a single parameter in the biophysical model (the ligand binding constant for the active state of LacI, $K_A$).

To see if this potential link between LANTERN and biophysics could be used in forward engineering, we attempted to use the LANTERN model results together with insight from the biophysical model to engineer improved inverted LacI variants. Most inverted LacI variants in the large-scale dataset have relatively high $EC_{50}$, and they are also somewhat leaky ($G_\infty > 1000$ MEF, compared with $G_0 = 158$ MEF for wild-type LacI). Based on the biophysical model, both $EC_{50}$ and $G_\infty$ of inverted variants can be reduced by decreasing the ligand binding constant for the active state, $K_A$, which tentatively corresponds to an increase in the second latent parameter of the LANTERN model. So, we chose three mutations with a significant predicted increase in that second latent parameter (S70R, V80L, and V136E). We synthesized and tested LacI variants composed of those mutations added onto the background sequences for two genetically distinct inverted variants. In both inverted backgrounds, the mutation V80L reduced $EC_{50}$ by a factor of 5 or 6, and reduced $G_\infty$ by a factor of about 1.3 (Fig 9, blue). The other two mutations, however, did not have the intended effect: S70R increased $EC_{50}$ in both inverted backgrounds (Fig 9, orange), and V136E resulted in constitutively high output (Fig 9, green). Although imperfect, this initial test of linking an interpretable, data-driven ML model to a biophysical model to engineer genetic sensors shows promise for engineering difficult-to-access phenotypes that differ significantly from the wild type.

## Discussion

We have demonstrated two approaches for precision engineering of genetic sensors and quantitatively evaluated their accuracy and the range of engineered phenotypes they can access.

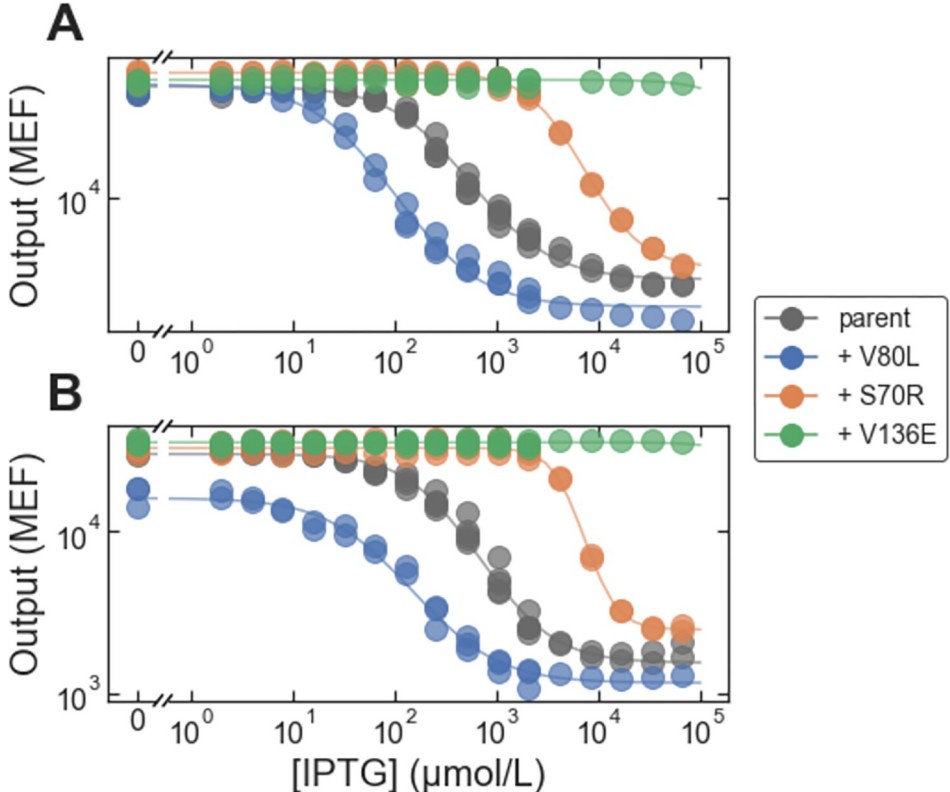

**Fig 9. Forward engineering to improve inverted sensors.** Each plot shows dose-response curves for a 'parent' inverted LacI variant and for that parent with the addition of mutations chosen to improve the inverted variant (by reducing $EC_{50}$ and $G_\infty$). (A) The parent variant has three missense mutations: A87P, V301M, and E357G. (B) The parent variant has five missense mutations: V96E, T154I, S158R, V238D, M254I, and V264I.

With *in silico* selection, we engineered sensors with $EC_{50}$ values spanning nearly three orders of magnitude with high precision (1.3-fold). In addition, we demonstrated that *in silico* selection can be used for facile, multi-objective engineering to give genetic sensors with specified values for both $EC_{50}$ and $G_\infty$, and with high accuracy relative to pre-defined specifications (1.22-fold and 1.14-fold for $EC_{50}$ and $G_\infty$, respectively). We also showed that *in silico* selection can be used for multi-objective engineering of more difficult and rare phenotypes: inverted sensors with specified $EC_{50}$, though with lower accuracy due to the relative sparsity of inverted variants in the large-scale dataset (1.6-fold to 2.6-fold for $EC_{50}$). With ML-enabled forward engineering we demonstrated that an ML model can be trained with a large-scale genotype-phenotype landscape dataset, and that model can then be used to predict the dose-response of new mutation combinations, again with good accuracy (1.3-fold to 1.9-fold for $EC_{50}$ and ~1.2-fold for $G_\infty$). We further demonstrated that an interpretable ML model can be used together with insight from a more explicit biophysical model to engineer inverted genetic sensors with improved $EC_{50}$ and $G_\infty$. To get a baseline for comparison of the performance of the precision engineering approaches, we measured multiple replicate dose-response curves for wild-type LacI (two biological replicates, with a total of 15 technical replicates measured on six different days). Across those wild-type replicates, the geometric standard deviation was 1.16-fold, 1.22-fold, and 1.11-fold, for $EC_{50}$, $G_0$, and $G_\infty$, respectively.

For both approaches to precision engineering, it is important that the large-scale dataset contains sequence variants with multiple mutations, i.e., not just data for variants with single

amino acid substitutions. Similarly, the dataset must contain results specifically related to each variant in the measured library rather than just an enrichment score associated with each mutation. With *in silico* selection, if we restrict the dataset to only single-mutant variants, the expected probability for success (i.e., engineering a dose-response satisfying the specification) drops significantly (S1 Appendix). Also, there are no single-mutant variants in the dataset expected to satisfy the specifications farthest from the wild-type (inverted dose response; or $G_\infty = 16$ kMEF and $EC_{50} = 10$ µmol/L or 30 µmol/L; S1 Appendix). So, with only single mutations, the range of phenotypes that can be engineered becomes more limited. Multi-mutant variants are also important for training the ML model, since multi-mutant data are required to make predictions for new mutation combinations without strong assumptions about the additivity and linearity of mutational effects [45].

To compare the accuracy demonstrated here with previous work, we are only able to find four examples of quantitative evaluation of predicted vs. measured genetic sensor dose-response. Two of those were for RNA-based sensors, and the other two were focused on engineering the dose-response of protein-based genetic sensors by varying the sequence of the cognate DNA operator (while using the wild-type protein sequences). Those previous publications included quantitative results for $G_0$ and $G_\infty$ (or the ratio $G_\infty/G_0$), and one included results for $G(L)$, but none of them included quantitative results for $EC_{50}$. Borujeni et al. developed a biophysical modeling approach to engineer RNA-based genetic sensors [13]. They tested the accuracy of the model by measuring the response of 67 riboswitches and showed that their model could predict the activation ratio, $G_\infty/G_0$, with approximately 2.5-fold accuracy (i.e., within 2-fold of the correct value for 55% of the tested riboswitches). However, their model was less accurate for calculating the values of $G_0$ and $G_\infty$ rather than their ratio (~8-fold and ~6-fold accuracy respectively). Angenent-Mari et al. trained several deep neural network models using a large-scale genotype-phenotype dataset for RNA toehold switches [14]. Their best model was able to predict $G_0$ and $G_\infty$ with about 3-fold accuracy. Yu et al. developed a biophysical model to predict how changes in promoter architecture and sequence affect $G_0$ and $G_\infty$ [46]. Their model was able to predict $G_0$ and $G_\infty$ with 1.6-fold accuracy across a set of 8269 designed *lac* operators (i.e., predictions within 2-fold of the true value 87% of the time). Zhou et al. used dose-response measurements for protein-based genetic sensors with 2632 combinatorially designed operator sequences to train regression models for $G(L)$ at each ligand concentration ($L$). Their best model had a predictive accuracy of about 1.2-fold [47]. By comparison, in our demonstration of the *in silico* selection method, all 16 of the engineered sensors with data shown in Fig 3 had both $EC_{50}$ and $G_\infty$ within 2-fold of the specified target values, and two of the three inverted sensors (Fig 4) had $EC_{50}$ within 2-fold or the target value. Also, our data-driven ML model was able to correctly predict $EC_{50}$ and $G_\infty$ within 2-fold for 76% and 97% of the tested LacI variants, respectively.

If we broaden our comparisons to include predictive models for constitutive gene expression, the best-known examples are probably the various models for predicting the translation initiation rate from ribosomal binding site (RBS) sequences [48–53]. In a recent evaluation of several of those models using data for nearly 10,000 RBS sequences, the models' predictive accuracy ranged from approximately 1.85-fold to 11-fold (between 23% and 74% predicted within 2-fold of the measured value), with the most recent iteration of the RBS calculator giving the best performance [54]. A biophysics-based model was also demonstrated for terminator strength in *E. coli*, with approximately 3.9-fold accuracy across a set of 582 natural and synthetic designed terminators [55]. More recently, LaFleur et al. developed a biophysical model for the strength of promoters in *E. coli* [56]. That model was able to correctly predict *in vitro* transcription rates with 1.6-fold accuracy across a set of 5388 designed promoters (i.e., within 2-fold of the correct value 92% of the time), though it was less accurate for *in vivo*

systems (approximately 2-fold accuracy). Similar predictive models of promoter function have been developed for eukaryotic cells [57–60]. However, those reports only evaluated model performance using the correlation coefficient, and the data comparing predicted and measured results are not available as part of the reports' data supplements. So, it is not possible to estimate the predictive fold-accuracy of those models with the available information.

In summary, the precision engineering approaches described here have very good accuracy compared with previous quantitative results. The question of how accurate an engineering method would need to be will depend on specific applications. Beal et al. have estimated that a target accuracy of 1.5-fold would be sufficient for most applications requiring engineered genetic regulatory networks [27].

The use of interpretable ML modeling in conjunction with a biophysical model also has the potential to become a useful engineering approach, as demonstrated here for the engineering of improved inverted LacI variants. But more rigorous methods would be needed to link the latent parameters of the ML model to the biophysical parameters before that approach could be used for engineering with quantitative precision. An alternative would be to fit the large-scale dataset directly with a biophysical model, if an appropriate model is available. One outstanding problem is that estimation of biophysical parameters from phenotype measurements can be ambiguous [61,62]. A large-scale measurement approach, with measurements of many different multi-mutation combinations could help to overcome ambiguity, since it provides information on mutational effects across many different genetic backgrounds that can help resolve those ambiguities [63]. However, that kind of approach will probably prove much more challenging for protein-based genetic sensors, where the same change to the dose-response curve can be explained by changes to several different biophysical parameters as shown by Razo-Mejia et al. [43] and demonstrated in our experience fitting the large-scale LacI dataset with a LANTERN model as discussed above.

For most applications, there will be some shift in context between the large-scale measurement and the application (e.g., a change in strain, growth conditions, and/or the genes that are regulated by the sensor). Ultimately, successful use of the methods described here will depend on the ability to predict how a genetic sensor's dose-response curve will change in response to those types of context shifts. The types of biophysical models discussed above, whether used in conjunction with interpretable ML or fit directly to data, provide a promising solution to the challenge of predicting function across different contexts. For example, Razo-Mejia et al. developed a biophysical model for allosteric regulation with LacI, and showed that it could accurately predict changes to the dose-response curve due to changes in LacI copy number or the interaction strength between LacI and its cognate operator [43]. Chure, Kaczmarek, and Phillips then demonstrated that the same model could accurately predict changes in the basal output level, $G_0$, due to cell growth at different temperatures and with different carbon sources [64]. Notably, Chure, Razo-Mejia, et al. showed that the model could also be used to predict changes in dose-response resulting from combinations of mutations (using single-mutant data) [41]. Although they did not include a quantitative evaluation of the accuracy of those predictions, it appears to be quite good (e.g., six of six predicted $EC_{50}$ within 2-fold of the correct value, based on a visual inspection of Fig 5A in [41]). Sochor showed that a similar biophysical model could be used to predict the *in vivo* dose-response curve of LacI using data from *in vitro* transcription measurements [65]. Finally, the model developed by LaFleur et al. [56] can predict changes in gene expression due to changes in sequence context upstream and downstream of a promoter site. So, although quantitative prediction of the effects of different biological contexts remains one of the outstanding challenges in the field [66], for genetic sensors at least, promising solutions exist. Admittedly, if biophysical models (or other means) are needed to correct for shifts in context between the large-scale measurement and the

application, that will add an additional layer of uncertainty in the use of the methods described here. But that just highlights the need for the best possible quantitative accuracy of the underlying large-scale measurements.

Currently, we are aware of only one large-scale dataset with quantitative results for the dose-response curves of a protein-based genetic sensor: the LacI dataset used here [28]. So, it is not yet possible to fully assess the generalizability of the methods presented here to other proteins. As an indication of the possible generalizability, though, we can compare the basic requirements of our methods with the requirements for directed evolution: both rely on the ability to generate phenotypic diversity via protein mutations. Directed evolution and related methods have been used to qualitatively improve a large variety of protein-based genetic sensors [17–26], in some cases with a single round of mutagenesis and a library diversity comparable to number of variants in the LacI dataset ($10^4$ to $10^5$ variants) [19–21,26]. Furthermore, in an approach similar to the *in silico* selection method described here, Ogawa et al. used deep mutational scanning data for a library of single-mutant XylS variants to identify mutations that alter the ligand specific of that protein-based genetic sensor [67]. So, as large-scale genotype-phenotype measurements become more accessible, we expect that the type of precision engineering approaches described here could be readily generalized to engineer different types of genetic sensors or other complex biological functions.

Compared with our approach, directed evolution has the advantage that it can be implemented with very large libraries of sensor protein variants: as many as $10^8$, compared with ~$10^5$ for the LacI dataset used here. So, we think that directed evolution methods will remain important for engineering new, hard-to-access protein functions, such as sensitivity to new ligands [6,10,68]. However, it would be very difficult to implement a directed evolution method for precision sensor engineering, for example to give a quantitatively specified $EC_{50}$. Similarly, promising new methods have been demonstrated for *de novo* computational design of genetic sensors [69], but those methods are unlikely to provide quantitative precision on their own. Therefore, we expect that methods like those described here will ultimate be used in conjunction with directed evolution or computational design, to provide quantitative precision when that is needed for real-world applications.

## Materials and methods

### Large-scale dataset

The large-scale dataset for LacI dose-response curves is described in ref [28]. It includes the estimated Hill equation parameters, $EC_{50}$, $G_0$ and $G_\infty$, for over 60,000 variants of the LacI genetic sensor, measured in *E. coli*. Those Hill equation parameter estimates, and their associated uncertainties, were obtained by fitting the measured dose-response curve of every variant to the Hill equation. That dataset is available via the NIST Science Data Portal, with the identifier ark:/88434/mds2-2259 (https://data.nist.gov/od/id/mds2-2259 or https://doi.org/10.18434/M32259). Here, we used the Hill equation parameter estimates and uncertainties as they are reported in that dataset.

### *In silico* selection

For the *in silico* selection results shown in Fig 3, LacI variants were chosen from the large-scale dataset based on the following criteria:

1. $EC_{50}$ within 1.2-fold of the target value (after correcting for systematic errors, see Fig 2C)

2. $G_\infty$ within 1.1-fold of the target value

3. $G_0 < 2$ kMEF

Those criteria were first applied using the median values reported in the dataset for $G_0$, $G_\infty$, and $EC_{50}$. That resulted in multiple LacI variants for each specification (between 18 and 1513). To identify the best variants to synthesize and test, the uncertainty information reported in the dataset was then used to estimate the probability for success of each variant: more specifically, the posterior samples reported in the dataset (from Bayesian estimation of the Hill equation parameters) were used to calculate the probability that each variant would meet the listed criteria. The variants were then ranked based on their probability of success; and the highest ranking three variants were selected for testing.

For the *in silico* selection results shown in Fig 4, a similar procedure was used to choose LacI variants, with the following criteria:

1. $EC_{50}$ within 1.5-fold of the target value

2. $G_\infty < 12.5$ kMEF

3. $19.2$ kMEF $< G_0 < 32.5$ kMEF

When applied to the median values for $G_0$, $G_\infty$, and $EC_{50}$, those criteria were only met by one or two LacI variants for each specification. Also, the calculated probability to meet the listed criteria was greater than 20% for only one variant per specification. So, only a single variant was selected for each specification.

## Strains, plasmids, and culture conditions

All reported measurements were completed using *E. coli* strain MG1655Δ*lac* [70], in which the lactose operon of *E. coli* strain MG1655 (ATCC #47076) was replaced with the bleomycin resistance gene from *Streptoalloteichus hindustanus* (*Shble*).

Dose-response curves were measured with flow cytometry using *E. coli* MG1655Δ*lac* transformed with variants of the pVER plasmid, described previously [28]. The plasmid contained different variants of the *lacI* coding DNA sequence (CDS), as described in the text, and an expression cassette with enhanced yellow fluorescent protein (eYFP) under the control of the lactose operator (*lacO*). The *lacI* CDS was verified with Sanger sequencing for each variant.

All cultures were grown in a rich M9 media (3 g/L $KH_2PO_4$, 6.78 g/L $Na_2HPO_4$, 0.5 g/L NaCl, 1 g/L $NH_4Cl$, 0.1 mmol/L $CaCl_2$, 2 mmol/L $MgSO_4$, 4% glycerol, and 20 g/L casamino acids) supplemented with 50 μg/mL kanamycin.

For flow cytometry measurements, *E. coli* cultures were grown in a laboratory automation system that included an automated liquid handler (Hamilton, STAR), an automated plate sealer (4titude, a4S), an automated de-sealer (Brooks, XPeel), and two multi-mode plate readers (BioTek, Neo2SM).

Cultures were grown in clear polystyrene 96-well plates with 1.1 mL square wells (4titude, 4ti-0255). The culture volume per well was 0.5 mL. Before incubation, each 96-well growth plate was sealed by the automated plate sealer with a gas permeable membrane (4titude, 4ti-0598). Growth plates were incubated in one of the multi-mode plate readers at 37°C with a 1°C gradient applied from the bottom to the top of the incubation chamber to minimize condensation on the inside of the membrane. The plate readers were set for double-orbital shaking at 807 cycles per minute. Optical density at 600 nm (OD600) was measured every 5 minutes during incubation, with continuous shaking applied between measurements (optical density at 700 nm and YFP fluorescence were also measured every 5 minutes). After incubation, the automated de-sealer was used to remove the gas-permeable membrane from each 96-well plate to enable automated passaging of cultures and sample preparation for flow cytometry measurements.

For each measurement, starter cultures were prepared from glycerol freezer stock in 5 mL of rich M9 media in a 14 mL snap-cap culture tubes. Starter cultures were incubated at 37˚C with orbital shaking at 300 rpm for between 4 h and 24 h prior to loading the automation system. The automation system then prepared 96-well growth plates, sealed and de-sealed the growth plates, incubated the growth plates, and prepared flow cytometry sample plates. The automated culture protocol consisted of the following steps:

1. Prepare first growth plate, with 450 μL rich M9 media in each well.

2. Pipette 50 μL of starter culture into each well in rows B-G of the plate (leaving rows A and H blank).

    a. Use a *E. coli* containing a different lacI variant for each row.

3. Seal first growth plate with gas permeable membrane.

4. Incubate plate in plate reader for 12 h to 14 h.

    a. Grow to stationary to provide a reproducible starting point for each measurement.

5. Prepare second growth plate with 490 μL in each well.

    a. Dilution series of isopropyl-β-D-thiogalactopyranoside (IPTG): 11 columns of a 2-fold serial dilution gradient and one column with zero IPTG.

6. Ten minutes before the end of the incubation cycle for the first growth plate, move the second growth plate to a heated station set to 47˚C.

    a. Ten minutes at 47˚C will pre-warm the media in the plate to 37˚C.

7. De-seal the first growth plate (after completion of the stationary-phase incubation cycle).

8. Pipette 10 μL from each well in the first growth plate to the corresponding well in the second growth plate.

    a. 50-fold dilution; using a 96-channel pipetting head.

9. Seal second growth plate with gas permeable membrane.

10. Incubate second growth plate in plate reader for 160 minutes.

    a. Sufficient for approximately 10-fold increase in cell density or 3.3 doublings.

11. Prepare third growth plate with 450 μL in each well.

    a. Same dilution series as in second growth plate.

12. Ten minutes before the end of the incubation cycle for the second growth plate, move the third growth plate to a heated station set to 47˚C.

13. De-seal the second growth plate (after completion of the 160 minute incubation cycle).

14. Pipette 50 μL from each well in the second growth plate to the corresponding well in the third growth plate.

    a. 10-fold dilution; using a 96-channel pipetting head.

15. Seal third growth plate with gas permeable membrane.

16. Incubate third growth plate in plate reader for 160 minutes.

17.  Prepare flow cytometry sample plate (round-bottom 96-well plate, Falcon, 351177).

    a.  Each well in rows B-G: 195 μL 1x PBS with 170 μg/mL chloramphenicol (Fisher BioReagents, cat. #BP904-100).

    b.  Rows A and H: PBS blanks, focusing fluid blanks, and space for calibration bead sample

18.  At the end of the incubation cycle for the third growth plate, pipette 5 μL from each well to the corresponding well in the flow cytometry sample plate.

At the end of the automated culture protocol, the flow cytometry sample plate was transferred to the flow cytometry autosampler for measurement.

### Flow cytometry

Flow cytometry samples were measured with an Attune NxT flow cytometer equipped with a 96-well plate autosampler using a 488 nm excitation laser and a 530 nm ± 15 nm bandpass emission filter. Blank samples were measured with each batch of cell measurements, and an automated gating algorithm was used to discriminate cell events from non-cell events [71]. Fluorescence calibration beads (Spherotech, part no. RCP-30-20A) were also measured with each batch of samples to facilitate calibration of flow cytometry data to molecules of equivalent fluorescein (MEF) [72–74].

For each LacI variant, the dose-response curve was taken to be the geometric mean fluorescence from flow cytometry as a function of the IPTG concentration in the media of the third growth plate. For many variants, data from multiple measurements were used, e.g., from biological or technical replicates, or data across multiple, overlapping IPTG dilution series to extend the range of inducer concentrations. For some biological and/or technical replicates, the cytometry results differed significantly from the consensus results from other replicates (i.e., $G_\infty$ more than 1.25-fold different from the consensus value). Data for those outlier replicates were not used. The Hill equation parameters and their associated uncertainties were determined by fitting all of the non-outlier cytometry data for each variant to the Hill equation using Bayesian parameter estimation by Markov Chain Monte Carlo (MCMC) sampling with PyStan [75].

### LANTERN ML modeling

LANTERN was fit to the LacI dataset with methods described in Ref [29]. In this model, LANTERN learns to predict observed phenotypes $y \in R^D$ given a one-hot encoded form of the genotype $x \in \{0, 1\}^p$ in two key steps. First, the genotype is projected to a low dimensional space $z = Wx$, where $W \in R^{K \times p}$ and $K \ll p$. Second, LANTERN learns a smooth non-linear surface connecting this low dimensional space to observed phenotypes: $y = f(z)$. Both the matrix $W$ and function $f(z)$ are unknown parameters and are learned by LANTERN in the form of an approximate variational posterior [76].

To quantify the predictive uncertainty of the LANTERN model for individual variants, we approximated the posterior predictive distribution for each variant under the learned model. This was done by taking Monte Carlo draws from learned approximate posterior (fifty draws were taken for each variant). Then, the mean and standard deviation of these draws were used to summarize the posterior predictive interval, as shown in Fig 8.

### Supporting information

**S1 Appendix. Supplementary information.** This appendix includes a table listing the mutations used for ML-enabled forward engineering and a summary of the results for *in silico*

selection with only single-mutant data.
(PDF)

**S1 Table. Dose-response data (from flow cytometry) for all LacI variants.** Data is in comma-separated value (csv) format. Column definitions:

  variant: the name of the plasmid variant used for the cytometry measurements;

  mutation_codes: a list of amino acid substitutions relative to the wild-type LacI sequence;

  clone: identifier for biological replicates (i.e., replicate colonies picked after transformation);

  IPTG: the concentration of IPTG in µmol/L;

  geo_mean: the geometric mean of the YFP signal measured by cytometry;

  geo_mean_err: the estimated uncertainty (one standard deviation) of the geometric mean;

  date_plate: the date of the measurement and the number of the 96-well plate used for the measurement (used to distinguish technical replicates);

  row: the row within the 96-well plate used for the measurement (used to distinguish technical replicates).
(CSV)

**S2 Table. Hill equation fit results (using cytometry data) for all LacI variants.** Data is in comma-separated value (csv) format. Column definitions:

  variant: the name of the plasmid variant used for the cytometry measurements;

  mutation_codes: a list of amino acid substitutions relative to the wild-type LacI sequence;

  log_g0: the base-10 logarithm of $G_0$ (in MEF);

  log_g0_err: the estimated uncertainty of $\log_{10}(G_0)$;

  log_ginf: the base-10 logarithm of $G_\infty$ (in MEF);

  log_ginf_err: the estimated uncertainty of $\log_{10}(G_\infty)$;

  log_ec50: the base-10 logarithm of $EC_{50}$ (in µmol/L);

  log_ec50_err: the estimated uncertainty of $\log_{10}(EC_{50})$;

  n: the Hill coefficient, $n$;

  n_err: the estimated uncertainty of $n$.

All parameter values are given as the posterior mean and uncertainties are one standard deviation of the posterior distribution from the Bayesian parameter estimation.
(CSV)

## Acknowledgments

We would like to thank Elizabeth Strychalski, Samuel Schaffter, and Edward Eisenstein for thoughtful comments on the manuscript.

## Author Contributions

**Conceptualization:** Drew S. Tack, Peter D. Tonner, Abe Pressman, David Ross.

**Formal analysis:** Peter D. Tonner, Nathan D. Olson, David Ross.

**Investigation:** Drew S. Tack, Peter D. Tonner, Nina Alperovich, Olga Vasilyeva, David Ross.

**Methodology:** Drew S. Tack, Peter D. Tonner, Sasha F. Levy, Eugenia F. Romantseva, Nina Alperovich, David Ross.

**Project administration:** David Ross.

**Software:** Peter D. Tonner, Nathan D. Olson, Eugenia F. Romantseva.

**Writing – original draft:** Drew S. Tack, Peter D. Tonner, Abe Pressman, Nathan D. Olson, Sasha F. Levy, Eugenia F. Romantseva, David Ross.

**Writing – review & editing:** Drew S. Tack, Peter D. Tonner, Abe Pressman, David Ross.

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
