## [Decision Letter · Decision Letter 0]

20 Jan 2023

PONE-D-22-31309Precision engineering of biological function with large-scale measurements and machine learningPLOS ONE

Dear Dr. Ross,

Thank you for submitting your manuscript to PLOS ONE. After careful consideration, we feel that it has merit but does not fully meet PLOS ONE’s publication criteria as it currently stands. Therefore, we invite you to submit a revised version of the manuscript that addresses the points raised during the review process. Please submit your revised manuscript by Mar 06 2023 11:59PM. If you will need more time than this to complete your revisions, please reply to this message or contact the journal office at plosone@plos.org. Please include the following items when submitting your revised manuscript:A rebuttal letter that responds to each point raised by the academic editor and reviewer(s). You should upload this letter as a separate file labeled 'Response to Reviewers'.A marked-up copy of your manuscript that highlights changes made to the original version. You should upload this as a separate file labeled 'Revised Manuscript with Track Changes'.An unmarked version of your revised paper without tracked changes. You should upload this as a separate file labeled 'Manuscript'.If applicable, we recommend that you deposit your laboratory protocols in protocols.io to enhance the reproducibility of your results. Protocols.io assigns your protocol its own identifier (DOI) so that it can be cited independently in the future. For instructions see: https://journals.plos.org/plosone/s/submission-guidelines#loc-laboratory-protocols. Additionally, PLOS ONE offers an option for publishing peer-reviewed Lab Protocol articles, which describe protocols hosted on protocols.io. Read more information on sharing protocols at https://plos.org/protocols?utm_medium=editorial-email&utm_source=authorletters&utm_campaign=protocols.

We look forward to receiving your revised manuscript.

Kind regards,

Hari S. Misra

Academic Editor

PLOS ONE

Journal Requirements:

 "HHS | National Institutes of Health (NIH):Sasha F Levy R01 HG011676; HHS | National Institutes of Health (NIH):Sasha F Levy R01 AI164530"

    "We would like to thank Elizabeth Strychalski, Samuel Schaffter, and Edward Eisenstein for thoughtful comments on the manuscript. S.F.L. is supported by NIH grants R01 HG011676 and R01 AI164530."

  "HHS | National Institutes of Health (NIH):Sasha F Levy R01 HG011676; HHS | National Institutes of Health (NIH):Sasha F Levy R01 AI164530"

Additional Editor Comments:

The manuscript has been reviewed by 2 subject experts. Both have appreciated work. I agree with them. Reviewer 1 has made important observation and has suggested substantial revision. A thorough revision of manuscript would be needed before it can be considered for publication.

Reviewers' comments:

Reviewer's Responses to Questions

**Comments to the Author**

1. Is the manuscript technically sound, and do the data support the conclusions?

Reviewer #1: Yes

Reviewer #2: Yes

2. Has the statistical analysis been performed appropriately and rigorously? 

Reviewer #1: No

Reviewer #2: Yes

3. Have the authors made all data underlying the findings in their manuscript fully available?

Reviewer #1: Yes

Reviewer #2: Yes

4. Is the manuscript presented in an intelligible fashion and written in standard English?

Reviewer #1: Yes

Reviewer #2: Yes

5. Review Comments to the Author

Reviewer #1: Review

This draft proposes two approaches for forward-design of genetically encoded biosensors, in silico selection and machine learning approach. Both methods rely on a large-scale dataset. The 'in silico selection' approach is to do a look up table search to find variants meet one or more parameter requirements. The second approach predict phenotypes of new variants after training a machine learning on the large-scale dataset LacI sequences. It’s also interesting the draft indicates some connection between machine learning model and biophysical model which might help better engineer genetic sensors. Overall, I think it is well organized paper. However, the statistical analysis parts are lack of details and some discussions in the paper needs more clarification. 1. Line 110 to 115. A system error between the cytometry measurements and the large-scale dataset is stated. And a best-case fold-accuracy after correction is reported. Is this correction applying to all the accuracy reported for different methods, if yes, please clearly states that, or it is interesting to explain why this only applies to this part.

2. Line 110 to 115 and line 136 to 139, it seems the approach get even better accuracy results on multiple objective specifications compare with single objective specifications. Is it a little weird to be more accurate with more complicate specifications? Please make some clarifications.

3. Line 263 to 285, the accuracy of previous works is discussed. To my understanding, these works are with quite different data set. If that’s true, it might be not directly comparable between these accuracy numbers.

4. Line 216 to 231, the evidence shows by experiment results may not be strong enough to indicate potential link between LANTERN and biophysics models. Consider the test size results, it might hardly can find any statistical significance supporting the combine use of machine learning model here and biophysics models.

5. Line 165 to 181, it might be good to show some cross validation performance on the LANTERN model train with this dataset, which can show overall fitting of model with this data set before test on new mutations.

6. It is also a bit ambiguous how easily techniques in this paper could be generalized to other genetic sensers, consider currently there is only one large-scale dataset with quantitative results for the dose-response curves of a protein-based genetic sensor: the Laci dataset used here, as stated by the author.

Reviewer #2: Present manuscript details two methods for precision engineering of genetic sensors. This paper proposes two approaches for design of sensors. In this revised version authors have made substantial corrections and answered most of the queries raised by the reviewers. Therefore, in my point of view this manuscript can now be accepted.

6. PLOS authors have the option to publish the peer review history of their article (what does this mean?). If published, this will include your full peer review and any attached files.

Reviewer #1: **Yes: **Qianshun Cheng

Reviewer #2: No

---

## [Author Response · Author response to Decision Letter 0]

6 Mar 2023

We thank the reviewers for their feedback on the manuscript. Or point-by-point response to their comments are below.

Reviewer #1: Review

This draft proposes two approaches for forward-design of genetically encoded biosensors, in silico selection and machine learning approach. Both methods rely on a large-scale dataset. The 'in silico selection' approach is to do a look up table search to find variants meet one or more parameter requirements. The second approach predict phenotypes of new variants after training a machine learning on the large-scale dataset LacI sequences. It’s also interesting the draft indicates some connection between machine learning model and biophysical model which might help better engineer genetic sensors. Overall, I think it is well organized paper. However, the statistical analysis parts are lack of details and some discussions in the paper needs more clarification. 

1. Line 110 to 115. A system error between the cytometry measurements and the large-scale dataset is stated. And a best-case fold-accuracy after correction is reported. Is this correction applying to all the accuracy reported for different methods, if yes, please clearly states that, or it is interesting to explain why this only applies to this part.

We intended to apply the correction to all of the subsequent results. However, when we rechecked the data analysis code, we realized that we had forgotten to apply this correction to the inverted sensor results. So, for consistency, we re-did the analysis applying the EC50 correction with the inverted variants. There were no significant changes in the results – since the correction is much smaller than the EC50 measurement uncertainties for the inverted variants. There was a slight change in the details of the in silico selection results: the calculated “probability of having an EC50 within 1.5-fold of the targeted value” increased slightly for some of the variants. For one variant that we did not select for flow cytometry verification, that probability increased from just below 10% to just above 10%. So, in the text, we increased the stated cutoff probability used for selecting the variants for flow cytometry verification from 10% to 20%. I.e., on lines 149 and 150 of the revised manuscript, “… for each target specification, there was only a single sequence with a greater than 10% 20% probability of having an EC50 within 1.5-fold of the targeted value…”

To make it clear that we applied the EC50 correction to all of the in silico selection results, we added the following sentence to the end of the paragraph where that correction is described: “For the subsequent evaluations of in silico selection described below, we continued to apply this correction to identify variants with EC50 values satisfying quantitative specifications.” We also added reminders that the correction was used in each subsequent paragraph where appropriate.

We thank the reviewer for identifying this ambiguity and believe the manuscript is improved with the implemented changes. 

2. Line 110 to 115 and line 136 to 139, it seems the approach get even better accuracy results on multiple objective specifications compare with single objective specifications. Is it a little weird to be more accurate with more complicate specifications? Please make some clarifications.

The calculation of the fold-accuracy is similar to the calculation of a standard deviation. Since that calculation is based on a finite number of data points (16 in this case), it has an associated uncertainty (i.e., the uncertainty of the uncertainty, which is almost never mentioned in any scientific papers). If we assume that the measurement errors are normally distributed (best case), the quantity analogous to the standard deviation is the logarithm of the fold-accuracy, and the unbiased estimate of the uncertainty in that standard deviation is SE(σ) = σ/sqrt(2*n-2), where σ is the standard deviation and n is the number of data points (16). With a fold-accuracy of 1.22, σ = log10(1.22) = 0.086, and the uncertainty in σ, SE(σ) = 0.086/sqrt(30) = 0.016. With a fold-accuracy of 1.31, σ_0 = 0.117. And, (σ_0 – σ)/SE(σ) = 1.9. So, the difference is probably not significant, given the finite number of data points. We added a short note indicating this is the revised manuscript.

3. Line 263 to 285, the accuracy of previous works is discussed. To my understanding, these works are with quite different data set. If that’s true, it might be not directly comparable between these accuracy numbers.

We agree that the previous works we compare to are a bit different than our work, but they are the closest comparisons we could find. If the reviewer is aware of other previous results that would be a better comparison, we would be happy to consider adding them.

4. Line 216 to 231, the evidence shows by experiment results may not be strong enough to indicate potential link between LANTERN and biophysics models. Consider the test size results, it might hardly can find any statistical significance supporting the combine use of machine learning model here and biophysics models.

We agree that the results in that section of the manuscript are not as quantitatively and statistically rigorous as the other sections. However, we think it is still a (marginally) valuable contribution, in that it suggests a possible route to predictive engineering of more difficult-to-achieve sensor phenotypes. Also, since we are completely transparent in describing what we did (e.g., including the things we tried that did not work), we don’t think there is any danger of a reader over-interpreting this result. So, we appreciate and agree with the reviewer’s concern, but we would like to leave this section unchanged.

5. Line 165 to 181, it might be good to show some cross validation performance on the LANTERN model train with this dataset, which can show overall fitting of model with this data set before test on new mutations.

We agree that cross validation would improve the impact of this section. This cross validation was shown in the previous publication describing the LANTERN modeling approach. We added a statement pointing to that cross validation in the previous publication.

6. It is also a bit ambiguous how easily techniques in this paper could be generalized to other genetic sensers, consider currently there is only one large-scale dataset with quantitative results for the dose-response curves of a protein-based genetic sensor: the Laci dataset used here, as stated by the author.

We agree that there is some uncertainty about how well the techniques will generalize to other systems. The same could be said for any other new techniques at the time of their first demonstration. However, based on the arguments given in the manuscript (lines 351-364 in the revision), we believe the techniques will generalize quite readily. 

Reviewer #2: Present manuscript details two methods for precision engineering of genetic sensors. This paper proposes two approaches for design of sensors. In this revised version authors have made substantial corrections and answered most of the queries raised by the reviewers. Therefore, in my point of view this manuscript can now be accepted.

---

## [Editor Report · Decision Letter 1]

13 Mar 2023

Precision engineering of biological function with large-scale measurements and machine learning

PONE-D-22-31309R1

Dear Dr. Ross,

We’re pleased to inform you that your manuscript has been judged scientifically suitable for publication and will be formally accepted for publication once it meets all outstanding technical requirements.

Kind regards,

Hari S. Misra

Academic Editor

PLOS ONE

Reviewers' comments:

Revised manuscript has improved, and reviewers' concerns have been addressed.

---

## [Editor Report · Acceptance letter]

20 Mar 2023

PONE-D-22-31309R1 

Precision engineering of biological function with large-scale measurements and machine learning 

Dear Dr. Ross:

I'm pleased to inform you that your manuscript has been deemed suitable for publication in PLOS ONE. Congratulations! Your manuscript is now with our production department. 

Kind regards, 

on behalf of

Professor Hari S. Misra 

Academic Editor

PLOS ONE